# Developing and Implementing an Action Plan among the “Orang Asali” Minority in Southernmost Thailand for Equitable Accessibility to Public Health Care and Public Services Following the United Nations Sustainable Development Goals

**DOI:** 10.3390/ijerph20065018

**Published:** 2023-03-12

**Authors:** Praves Meedsen, Chutarat Sathirapanya

**Affiliations:** 1Institute for Development of Civil Officials in Southern Border Provinces, Southern Border Provinces Administrative Center, Meung, Yala 95000, Thailand; 2Department of Family and Preventive Medicine, Faculty of Medicine, Prince of Songkla University, Hat Yai, Songkhla 90110, Thailand

**Keywords:** Orang Asali, minority, universal health coverage, health care, willingness

## Abstract

Ending social inequality by 2030 is a goal of the United Nations’ endorsed sustainable development agenda. Minority or marginalized people are susceptible to social inequality. This action research qualitatively evaluated the requirements for and barriers to full access to public services of the Orang Asali (OA), a minority people living in the Narathiwas province in southernmost Thailand. With the cooperation of the staff of the Southern Border Provinces Administrative Center (SBPAC), we interviewed the OA, local governmental officers and Thai community leaders regarding the OA’s living conditions and health status. Then, an action plan was developed and implemented to raise their living standards with minimal disruption to their traditional cultural beliefs and lifestyle. For systematic follow-ups, a Thai nationality registration process was carried out before the assistance was provided. Living conditions and livelihood opportunities, health care and education were the main targets of the action plan. Universal health coverage (UHC), according to Thai health policy, was applied to OA for holistic health care. The OA were satisfied with the assistance provided to them. While filling the gap of social inequality for the OA is urgent, a balance between the modern and traditional living styles should be carefully considered.

## 1. Introduction

For better well-being for all and at all ages by 2030, the “Sustainable Development Goals (SDGs)” agenda was endorsed by the United Nations in 2015. No poverty (Goal 1), good health and well-being (Goal 3) and quality education (Goal 4) were highlighted among the seventeen goals [1]. Poverty and illiteracy are common causes that limit individuals’ accessibility to public health care, leading to poor hygiene and illness. Additionally, poverty, illiteracy and illness have been considered factors of a vicious cycle of unhealthy lives. Therefore, providing equitable access to health care along with quality education and raising economic standards should be strategies for improving the well-being of individuals in a society. In Thailand, the national “Universal Health Coverage (UHC)” payment scheme for the health service was launched in 2002, so that all Thais could access the public health care system equitably from birth until death. The UHC is composed of various aspects that complete the health service loop, i.e., health promotion, prevention, treatment and rehabilitation, which aim at healthy lives for all in the country. Moreover, other public services necessary for daily living (i.e., electricity, piped water and sanitation), standard education, occupational training, employment assistance, etc., should be available equitably among the people of a nation. However, there are gaps in the nationwide coverage of the UHC program as well as other public facilities in Thailand, especially among minority groups or marginalized peoples, migrant workers, nomads, etc. Inaccessibility to the public health care system and a lack of health knowledge among these people to manage their own health cause them to be highly vulnerable to unhealthy living conditions and to acquire either communicable or non-communicable diseases (NCDs), most of which are preventable. Providing accessible health services and other facilities necessary for better living conditions to these special groups of people should be systematically organized and accepted by these target groups of people.

The “Orang Asali” (OA), which means “the first people” or” original people”, is a group of indigenous people who have lived dispersedly throughout the Malay peninsula and southernmost Thailand, where it bordered with Malaysia, for 25,000 to 60,000 years [2,3]. According to anthropological information, the OA comprise three ethnically different groups, i.e., the Senoi, Porto-Malay (or Aborigine Malay) and Negrito [4]. The Senoi are the largest group of OA in Malaysia, while the OA in Thailand are mostly Negrito. In Thailand, the OA people live primarily in two mountain ranges, the Banthut mountain range in the Phattalung, Satun, and Trang provinces and the Sunkalakiri mountain range in the Yala and Narathiwas provinces. The latter group is close to the Thai–Malaysia border; therefore, their cultural beliefs and lifestyles are similar to those of the OA in Malaysia. The available research related to the OA in Thailand to date mainly involve anthropology, social status, and lifestyle rather than living conditions, health care services and education. According to a systematic review, the OA in Malaysia commonly suffered from malnutrition, lower growth rates in children, soil-transmitted helminths, pulmonary diseases and cardiometabolic diseases [5]. These acquired diseases also occur among the OA in Thailand due to similar living conditions and environments. To promote secure and healthy living conditions among the OA in Thailand, various interventions have been applied, such as providing these groups with Thai citizenship and accessibility to standard education, especially for school-aged children; health education and health promotion programs; and resettlement areas prepared for their permanent residence. According to the policy of the Ministry of the Interior of Thailand (MOI-T), OA in Thailand are considered Thai citizens, as other Thais are. Numerous operational plans have been deployed to improve their living conditions and increase their ability to access all public services and facilities.

In the public health sector, the provision of equitable access to public health services is an aim of health care system management for the OA based on the respect for their human rights, as for other Thais, and in response to one of the SDGs. The UHC is a key measure to eliminate inequitable accessibility to health care service [6]. For some time, the OA in the study area have received basic living support, including medical care from the local government when the requested. We expect that the current registration program for the OA as Thais will solidify a sustainable life-supporting system for them. The registered OA would then be able to access Thai public services, as other Thai citizens are able to. Furthermore, the outcomes and progression for the improvement of their living conditions can be followed systemically. Herein, we describe a pilot program of actions to promote the OA’s voluntary adoption of public services and facilities, including public health care services among a group of OA living in the Chanae district of Narathiwas province.

## 2. Material and Methods

### 2.1. Study Population and Setting

This was a qualitative study using semi-structured interviews for data acquisition. The study participants were nine OA leaders and their representatives selected by the OA villagers from Toapaku and Biyis villages (OA); five local governmental personnel working in the offices of civil registration, education, agriculture and public health (GP); and six local Thai community leaders or associate leaders (TC), e.g., the heads of Thai local villages, the head of sub-district offices, religious leaders, etc. We selected two of five OA villages in the area in which the villagers had settled permanently as the study sites. Two weeks after the research information, objectives and process were clearly described to the OA villagers via translators, verbal consents were obtained voluntarily from the OA villagers because they are not able to understand spoken or written Thai. Additionally, the GP and TC groups were informed of the study process and written consents were obtained. The whole research study process was conducted in the OA settlement areas and the local governmental offices in Chanae District, Narathiwas province, with the support of the Southern Border Provinces Administrative Center (SBPAC), which acted as the coordinator for the related local governmental agencies.

### 2.2. Preparation for Data Collection

After ethical approval and consent from the study participants was obtained, the research team started to gather preliminary information regarding the residential locations and the environments of the two study OA villages, their usual lifestyle and, significantly, their willingness to adopt Thai citizenship. On the official side, the relevant Thai laws or regulations and practical guidelines for verification of the OA as Thai people were reviewed. The preliminary information included an initial interview with the GP and TC groups regarding what had been carried out previously and the associated outcomes. Then, the in-depth interview questions were designed and tested for content validity by three experts in anthropology, public health and qualitative research. The questions were divided into three sections according to the research participants, i.e., the OA, GP and TC groups. We used translators who understood the OA spoken language as assistants in data collection. We visited and interviewed the OA participants in their homes, where we spent an average of two hours to complete each interview. We spent an average of one hour to interview the GP and TC participants at their offices. The interviewed content was recorded on audio recorders for later review and validation.

The questions used for the interviews with the study participants were as follows:

For the OA:
○After receiving the information from local Thai officers, are you willing to adopt Thai citizenship and why?○After receiving the information from local Thai officers, do you understand your rights and responsibilities after you become Thai.○In the past, how did you receive information about the registration process?○How do you contact other persons who are not OA like you?○What are the activities in your daily life?○Have there been any recent changes to your current living conditions, and if so, what are they?○How do you regularly manage your living areas, food supply and treatment for sick persons?○What is your perception of how your life and the life of your community might change if you accept Thai nationality?○Do you currently access any Thai official services, such as livelihood assistance, education and health care?

For GP:
○How many groups of OA are currently living in the Chanae district?○How many families are there in each group, and how many people per family?○What are their houses (“Tub” in Thai) built of?○In what kinds of environmental conditions do they usually build their houses?○What means do Thai officials use to contact the OA and who is/are the mediator/s helping the contacts?○What will be the steps for providing Thai identification cards to the OA?○What actions or services will the Thai governmental agencies provide for the OA, e.g., livelihood assistance, education or health care, after they receive Thai citizenship?○How do you expect an OA to perceive the changes in living conditions or services that will be provided to them after receiving Thai citizenship?

For TC:
○What kinds of settlements (permanent or migratory) do they (OA) have?○How long have they settled here?○What are their houses (“Tub” in Thai) built of?○In what kinds of environmental conditions do they usually build their houses?○What harvestable natural products do they use or consume, and how do they harvest them?○What were the previous living conditions of the OA who are going to be registered as Thai citizens?○What factors do you think will encourage an OA individual to accept the official offer to be registered as Thai?


### 2.3. Data Analysis

Before we started the analysis, the GP and TC reviewed and validated the in-depth interview content themselves, while an independent translator was used to ensure the correctness of the translation of the OA’s interview content. We performed data analysis following the “Thematic Analysis” principle [7,8], which used six steps: (1) data familiarization and writing familiarization notes, (2) systematic data coding, (3) generating initial themes from coding and collated data, (4) developing and reviewing themes, (5) refining, defining and naming the themes, and (6) writing the report. After the analysis, an action plan was co-designed by the study team staff, GP and TC based on the results from the thematic analysis and the designed action plan was implemented among the OA study participants. We performed a short-term outcome evaluation, and long-term evaluation was also planned to be done in the future (Figure 1).

### 2.4. Ethical Considerations

Ethical approval for the study was granted by the Ethics Committee of Public Policy Institute, Prince of Songkla University (EC code: 008/64, date of approval 10/06/2021). We strictly followed the 1964 Declaration of Helsinki, its amendments and related guidelines for the ethical conduct of research studies. All the participants’ identifiable information was completely anonymized.

## 3. Results

### 3.1. Study Participants’ Characteristic

We enrolled nine OA leaders and their representatives selected by the villagers of the Toapaku and Biyis villages, five local governmental personnel (GP) and six local Thai community leaders or associates (TC) of the Chanae District, Narathiwas province. The characteristics of the study participants are shown in Table 1.

### 3.2. Preliminary Information from the Initial Survey

Initially, we interviewed the GP and TC groups on three topics for preliminary information before in-depth interviews and subsequent action planning were carried out. The aims of this preliminary interview were to evaluate the preparedness of the study participants from the Thai official sector, who would be involved in the process of planning further action, as well as the current Thai regulations. The topics of the preliminary interview included the following.

#### 3.2.1. The Settlement Areas and the Lives of the OA in the Study Area

Overall, there were five villages of OA in the study area, of which the villagers in three villages lived nomadically by hunting or harvesting natural forest products on the mountain, while the remaining two groups, i.e., the Toapaku and Biyis villages, had permanent settlements. The Toapaku village had six households with 32 members, while the Biyis village had five households with 27 members. The leaders of both villages were males whose leadership was derived from their ancestors. They spoke OA, Malay or, less frequently, the Thai language.

The preliminary information obtained from the talks with the GP and TC groups in the area included the following:


*“The OA have lived along the Dusongyor mountain range in Chanae district for at least 200–300 years. During the earlier days, the forests on the mountains were rich with many natural products adequate for their household use. They usually found a new place to settle when the former settlement place no longer supplied enough food and water for living.”*
[TCa1]


*“There are 2 groups of OA who reside permanently in this area with a total of 11 households and 59 members. They usually live in mountainous areas where a stream flows. Their houses, called “Tub” (in Thai), are simple and commonly built from bamboo and local forest woods. Their houses have a floor high off the ground and the roofs are made of a kind of palm leaves easily found in their living areas. Each house contains only one family. They prefer to live and move to a new settlement together in group.”*
[GPb1]


*“These two OA groups have settled in their current living places for at least 1 year. They usually harvest or hunt forest products or wild animals only enough for their consumption (not for commercial purposes) during the daytime and return home at night. This is not like in the past when they regularly moved to a new place every 9–10 days when the harvestable forest products became scarce.”*
[TCa2]


*“The building styles of the houses of those who settle permanently are different from the styles of those who migrate from place to place in the forest. The settled houses are built with more stability. They have high raised floors, stable wooden support poles and roofs. The “Tubs” of the migratory OA groups are simply built using bamboo, forest wood and leaves easily found in their living areas, and the houses have no walls. The building style of settled houses seems to indicate that they intend to stay permanently in this place.”*
[TCa1]

#### 3.2.2. Current Thai Law and Regulations and Previous Experience of Granting Thai Citizenship to OA in the Other Provinces

The district head governor and his staff followed the relevant policies of the MOI-T and studied the laws and regulations applicable to this issue. They set up legally based portals for the OA to receive Thai citizenship. Initially, the OA were clearly informed about the steps required to obtain Thai citizenship, and their rights and responsibilities after becoming Thai citizens according to the Thai laws. Significantly, it was emphasized to the OA that their acceptance of Thai citizenship was voluntary if they fulfilled the required legal criteria for becoming Thai. The local governor’s teams also studied a previous successful project of verification and providing Thai citizenship to OA carried out in the Betong district of the Yala province as a model.


*“We follow the policy of the MOI-T that OA in Thailand are regarded as Thai citizens. They have the same freedoms and rights to access government facilities and support as other Thais. Therefore, we try to conduct the process of granting Thai citizenship to the OA and have them legally registered as Thais according to their willingness.”*
[GPa3]


*“We studied the experience of the local government in Betong to deal with the status of the OA there and found that they applied the “Regulations for Civil Registration” of the Department of Governance, Ministry of Interior (5th revision, 2008) to handle any problems. In Betong, the local governor’s team traced an OA individual’s family tree in combination with the confirmations from local Thai witnesses that the OA had grown up and lived in the area for a long time before as criteria for the approval of Thai citizenship for the OA. Finally, the OA in Betong were registered into the Thai citizen list, and Thai identification cards (IDC) and numbers (IDN) were provided to them. We plan to use the practices in Betong as a model for our plan too.”*
[GPb2]


*“Historical and anthropological information confirm that the OA have settled in this area for many thousands of years. We observe their physical characteristic, living conditions, livelihoods and languages which are compatible with the original OA described in both Thai and western historical records to support their presence in southernmost Thailand.”*
[GPb1]

#### 3.2.3. Evaluation of the OA’s Willingness to Adopt Thai Citizenship and the Barriers of the Verification Process for the OA as Thais

Two months prior to our interview, the district governor’s team staff evaluated the OA’s willingness to adopt Thai nationality by informing them of their rights and responsibilities as Thais before they could decide freely. At the same time, they prepared the OA for our research team to perform the in-depth interviews for data collection.


*“We informed the OA participants about the study and asked them about their willingness to adopt Thai citizenship among those who lived in permanent settlements first as they were easy to contact and knew much about the local Thai living conditions and cultural practices. For the OA who regularly migrate for resettlement on the mountain range, it was very difficult to get in touch with them.”*
[GPa1]


*“Language was a significant barrier of providing clear information to the OA. We assumed that there were some possible misunderstandings about the citizenship registration process and the answers of their willingness to be registered as Thais between the district governor’s staff and the OA. Hence, we had to contact the OA via some persons that they relied on and understood the OA language well.”*
[GPb1]


*“The OA in this area rely on their employers who have hired them for casual labor for a long time. Apart from the employers, Thai villager leaders or their assistants who usually support the OA can communicate with them well. We asked these persons to help the district governor’s team by contacting and making an appointment with the OA before our visit to explain the details of the information to the OA. Every sector of the district government team including representatives of Thai civil registration, district land management, agriculture, public health, education, etc., were prepared to respond to relevant queries from the OA.”*
[TCa1]

### 3.3. In-Depth Interview Results, the Developed Action Plan and Actions Performed

The research team, with the assistance of the local governor’s staff, visited the OA who lived permanently on the Dusongyor mountain to carry out in-depth interviews for data collection. We interviewed the OA regarding their living conditions and health care, education and other public services they required. We once again explained the rights and the responsibilities they would have after they were registered as Thai citizens before their willingness to accept the offer was confirmed. We first traced the evidence to confirm their longtime settlement and their relative links with other OA members living in this area. If the OA fulfilled these criteria according to the MOI-T’s policies and regulations, civil registration process as Thais was done; and an IDC and IDN were eventually given to them. The district government strictly followed the guidelines for the civil registration process issued by the MOI-T and strongly emphasized the preservation of the OA’s traditional ways of living despite their new nationality as Thai people.

#### 3.3.1. Visiting the OA Living Areas to Ask Their Willingness

We found that the villagers of the two villages were willing to be registered as Thais and to comply with Thai laws. They understood their rights as granted by Thai officials and the obligatory responsibilities to Thai society, whilst their traditional ways of living would be preserved. Their reasons for the adoption of Thai citizenship were that their children could attend school, they could receive health care and other public services, participate in health promotion programs, and receive an adequate food supply.

The medical treatment among the OA depended on ancestral practices. For example, every pregnant woman went through childbirth naturally with assistance from a village midwife without a prenatal evaluation of maternal and fetal risks. No vaccinations for newborns or the aged were provided. Although they had experience in the medicinal properties of many natural products used as medicines, many complicated diseases were unable to be treated successfully.


*“We would like to have adequate food for our kids as the harvestable forest products have progressively diminished over the years. On some days we catch or trap no wild animals to cook for our kids.”*
[OAa1]


*“The plants or wild animals in the forest reduced in number from previous years. Although we try to plant cassava trees to collect their roots, the harvestable products are smaller than before. So, we are necessary to settle in permanent living areas and become employees, instead.”*
[OAa3]


*“Sometimes, only one wild cock is caught for our food. So, we cook it for our kids first. The adults have to reduce or miss their meals and wait for other cooked foods.”*
[OAb1]


*“The Southern Border Provinces Administrative Center (SBPAC) has followed the living conditions of the OA for a long time. Because they live on high mountain ranges, travelling to their living areas to provide the information about various legal and social regulations or other necessary social or living supports is very troublesome. In the initial visits of the district governor’s staff, the OA reported their willingness to be registered as Thai.”*
[GPb2]


*“Their lifestyles (the OA) have changed after they became employees in Chanae district. They have learnt and understand about Thai citizen’s basic rights from their co-workers or the locals.”*
[TCb1]


*“We informed the OA that the Thai government regarded them as Thai people, and the Thai government would provide living facilities and financial support for their children, the aged and unemployed persons. The Universal Health Coverage program of Thailand would cover all the disease prevention and hospital treatment costs.”*
[GPa3]


*“The SBPAC has had an action plan to provide Thai citizenship for the OA since 2019. The actions planned are registration of the OA as Thai nationals, providing them with Thai IDCs and citizen IDNs as well as other welfare supports, e.g., UHC, monthly payments for the aged and newborns, etc., and improving their living conditions and livelihoods, children’s education, vocational training. All the actions are based on the principle of obtaining equitable living condition with other Thais and compliance with the current Thai regulations and laws, whilst their traditional lifestyles are preserved. We asked for their cooperation in forest preservation.”*
[GPa2]


*“Her Royal Highness Princess Mahachakri Sirindhorn gave an idea to us to help the OA to improve their living conditions while retaining their identity through their traditional livings and cultural practices. In cooperation with the “Supporting the Minority of Orang Asali Network”, the SBPAC facilitates the process of providing the OA with Thai nationality, welfare support, and improving the life skills necessary for modern living conditions.”*
[GPa1]


*“In our talks, we (the OA) recalled an event in which a man living in a village fell from a tree breaking his pelvic bone and required a treatment in hospital, but he had no Universal Health Coverage payment support to pay the treatment cost. So, we think it will be better for us to decide to be registered in the Thai civil registration list so that the Universal Health Coverage payment scheme is open to us, and the cost of treatment will be covered by this payment scheme when we are necessary to visit a hospital for treatment.”*
[OAb2]

#### 3.3.2. Clearance of Legal Issues and Preparation for Providing the OA with Thai Nationality

To save the expense and time travelling to the various district offices to complete the steps of verification and registration process by the OA themselves, the district governor’s team and associated local governmental agencies visited the OA settlements to complete the process, following which, the Thai IDCs and IDNs were given to them.


*“It is very troublesome for the OA to have them travel to the various district offices to complete the registration process. Our team and other district governmental officers will jointly visit them at their settlement sites again to trace the evidence of their long presence in this area, their relative relationships or a family tree based on the legal standards of the MOI-T before registering them as Thai people and adding their names to the Thai civil registration list.”*
[GPb2]


*“We will follow the regulations and legal guidelines for the registration process. When the registration process is completed, we will provide Thai IDCs to any OA age 7 years and older according to Thai law.”*
[GPb1]


*“These OA are voluntarily accepting registration as Thai citizens. Based on the policy of the MOI-T, the verification of long inhabitation in this area and relative relationships are confirmed by the local Thai community leaders, religious or social activity leaders, or their employers.”*
[GPa3]


*“We learnt from the successful registration of the OA in Betong, Yala province. In Betong, they traced the evidence that confirmed the long settlement of the OA in the area and their relative relationships before providing them with Thai citizenship. Our district governor’s team in association with other governmental support and service agencies will follow the same process carried out in Betong.”*
[GPa2]


*“In Betong, the local governmental team also added a program for improvement of living conditions to the registration process for the OA there. The operations yielded satisfactory outcomes to both the OA and the Thai local governmental officers.”*
[GPb2]


*“The OA who receive Thai nationality will have IDCs and IDNs starting with the number “5” at the beginning which means that they are a minority person or a foreigner who have been approved to be registered in the Thai civil registration list.”*
[GPa2]


*“The policy and practice guidelines launched by the MOI-T will facilitate the local district governor and associated governmental sectors to complete the registration process.”*
[GPa3]


*“We will provide the information regarding the rights that the OA will obtain from Thai government when they are completely registered as Thais, such as various financial and social welfare programs.”*
[GPb1]


*“The Thai government asks the OA for their cooperation in forest preservation, while their traditional ways of living will be preserved. The adoption of Thai citizenship will be voluntary depending on their own decisions. The OA who have settled in the national forest and wild animal preservation zones will be informed of the same practical principles before they can decide to adopt the offers freely. Alternatively, this group of OA can voluntarily resettle in one of the governmental resettlement areas provided for minority people.”*
[GPb2]

#### 3.3.3. Theme Development

After the completion of the study participant interviews, the themes were developed under the thematic analysis disciplines as follows:
**Themes****Codes**A. Preparation for the registration process for the OA1. A plan for improvement of the overall well-being of the OA is required.2. Respect their equitable human rights as Thai people.3. Compliance with legal standard practices for the provision of Thai citizenship to the OA4. Learning from the previous practices of the other governance areas.Living areas and livelihoods of the OA
Two of five OA villages were settled permanently, whilst the rest of remain nomadic.The harvestable forest products have progressively diminished over recent years, leading to inadequate resources to maintain adequate household consumption.Some OAs worked as laborers in the Chanae district, Narathiwas, to earn a living.OA children had malnourishment and illiteracy that affects their growth and development.The OA demand that they will be allowed to maintain their traditional ways of living with the forest after they are registered as Thai.Livelihood assistance, health care and all-level education suitable for each OA individual’s needs are necessary.The OA used ancestral methods to treat illnesses they experienced. They had no knowledge of health prevention or promotion.They were very anxious when they needed to visit a hospital due to their misunderstanding of current medical treatments.
Studying current Thai laws, regulations and national policies to facilitate the process of the provision of Thai citizenship to the OA by local governmental staff.
Because the OA are regarded as Thai people, the Ministry of the Interior, Thailand (MOI-T), instituted a policy for the registration of the OA as Thai citizens.Related acts and regulations, including guidelines for the verification of the Thai citizenship process will be reviewed and discussed among the governmental sectors at both the national and local levels.The local governmental agencies will collaborate in planning and carrying out the registration process.The successful registration process carried out in the Betong district of the Yala province was studied as a model.
The OA are willing to adopt Thai nationality.
Detailed information regarding the OA’s rights and responsibilities as Thai citizens was provided before their voluntary decision.Their ethnic identities and lifestyle will be preserved.Earning a living, health care access and basic or vocational education for children or adults, respectively, were considered essential for the OA.
B. Registration process according to the developed action plan1. The legal process of verification and registration as Thai people.2. Registered OA obtain equitable rights and have the same responsibilities as the other Thais.3. Collaborative work of related local government agencies is a key for successful registration.4. Living conditions, health service and education are the three main targets for the development of the OA’s well-being.5. Health services under the UHC payment scheme is essential for the OA to access health care.The local governmental staff visited the OA living area on the mountain to ask about their willingness to be registered as Thai people.The OA would like to accept the conditions after registration as Thai.The registration to accept Thai citizenship is voluntary.Personal verification will be carried out by local government staff based on the MOI-T’s policies and regulations.Language was an information provision barrier.The registration practice successfully carried out in the Betong district, Yala province, is to be followed.Thai IDCs and IDNs will be provided to OA aged 7 years or over, and they will be listed in the Thai civil list after completing the verification process.A parcel of land for residence or earning a living will be provided.Health insurance under the UHC payment scheme of the Thai public health system will be provided to the OA.UHC will support payment for the OA to receive health services.OA children will be allowed to attend public schools to study Thai.C. Outcome evaluation1. Immediate after-action evaluation of the registration process outcomes focusing on improved living conditions, health services accessibility and children’s education. 2. Long term follow-ups and repeated evaluations in the future are planned.The OA’s homes were redesigned for hygienic living.Follow-ups will be carried out to ensure the OA have received equitable social support and access to welfare programs as Thai citizens.Evaluation of the understanding of and accessibility to public health services under the UHC payment scheme among the OA and their satisfaction with the services will be evaluated.Teachers and local Thai community leaders will encourage OA children to attend a primary school.The OA parents and their children’ satisfaction with the organized education system will be evaluated.

#### 3.3.4. Official Provision of Thai Citizenship to the OA According to the Plan

Nearly 3 months after the survey of the OA’s willingness to be registered as Thai and the preparation by the district governmental agencies, the Chanae district governor’s team and the associated agencies managing the legal issues and planning the process for registration and provision of Thai citizenship, provided Thai IDCs and IDNs to the OAs.


*“In June 2022, the Thai citizenship for OAs was approved and Thai IDCs were provided to 25 and 20 OAs who aged from 7 years old from Toapaku and Biyis villages, respectively.”*
[GPb2]


*“On 31 August 2022, the Yala provincial governor presented the IDCs and the citizenship confirmation documents to the OA who had received the approval to be registered as Thais following the registration process. This signifies the OA have the rights to receive various governmental supports such as health services, education, and monthly financial support for the aged and newborns. The whole process was implemented to get rid of social inequality based on a Thai strategy for national development theme which states that “We will never leave anyone behind.”*
[GPa3]


*“The head of the Cooperation Center for Development and Special Activities under the Royal Initiations (CCDSR) under the SBPAC in cooperation with the Chanae district governor’s team, the head officer of the local forest protection office, Narathiwas province, a representative of the Department of Forestry, and representatives from the Sukirin Self-dependent Resettlement Area visited the OA settlement areas in the forest to help them improve their housing and livelihoods options.”*
[GPa1]


*“The SBPAC has facilitated the registration process for the OA so that they can access public facilities and other public supports like Thai people for raising their basic living standards in health, education, livelihoods, etc.”*
[GPb1]


*“The SBPAC is requesting a parcelof land from the Department of Land Management to establish a new resettlement area for the OA where they can grow crops and raise livestock for adequate household consumption. The project is under negotiation among the related agencies now.”*
[GPa2]


*“On 31 August 2022, we (the OA) received the IDCs and other related welfare cards. The community leader gave us the detailed information in the Malay language. We thank the district governor’s team for helping us to receive the IDCs and their visit to our living areas in village No. 7 (Moo 7, in Thai).”*
[TCb2]


*“The head of the Cooperation Center for Development and Special Activities under the Royal Initiations (CCDSR) had the representatives of the National Health Guarantee Office to travel with him to visit the OA. Additionally, the district public health officers and the associated team set up a field meeting for open discussion regarding the rights for the OA after registration to access public health care services according to item 18 (13) of the National Health Guarantee Act, 2002”*
[GPa2]


*“The representatives of the National Health Guarantee Office explained the details of the rights the OA would have after the registration to the OA leaders or representatives before they transferred the information to the individual OAs living on the mountain area.”*
[GPb2]

### 3.4. Post-Action Short-Term Evaluation

After the registration process for the OA was completed, we carried out a follow-up visit a few weeks later to assess how frequently the OA accessed public support, as well as their satisfication. We found that the OA were satisfied with the help of the local governmental agencies to improve their well-being and quality of life. Since they received their Thai citizenships, every individual OA was able to own a parcel of land to plant crops or raise livestock, access the public health care or health promotion services, and their children were able to attend the local primary schools, etc.


*“The first time we visited the hospital, we were very nervous and felt insecure. However, with the help of a local public health officer who took the injured OA to the hospital and the cooperation between the SBPAC and the hospital administration staff, the injured OA received the treatment free of charge. Initially, the patient was very concerned about the hip surgery advised by the doctor that the whole flesh on his body would be taken away during the operation.”*
[OAb2]


*“Teachers from the Out-School (Informal) Study Office and the village assistant leaders cooperatively set up study classes for the OA children to attend.”*
[GPb2]


*“The children enjoy learning in the classes. They usually attend their classes on time. We are slowly developing a study program for them. Asking the OA children about their future careers, they aimed to be soldiers or to have better lives like the local Thai people.”*
[GPa2]

Moreover, the informal education and vocational training were provided for OA who were 15 years or older. Scheduled classes were regularly organized in the community-shared building in which 10–20 OA youths attended each class. To assist in the improvement of their living conditions, they were taught to crop vegetables and raise livestock to maintain their food security. Additionally, seeds, baby chickens, baby ducks, etc., were provided.

Regarding public health services, thirteen and five OA individuals voluntarily received one and two doses of COVID-19 vaccine, respectively. They responded well to the COVID-19 vaccination campaign. They still preferred to use traditional herbal medicines for initial treatment, except for a complicated disease for which they were willing to receive modern medical treatment from the Thai community health volunteers, who regularly visited them, or in the district hospital when it was required. The number of OA utilizing modern medical services increased.

## 4. Discussion

One principal item in the Thai constitution emphasizes equitable rights of all Thais to access and receive public services or support. The OA in this study, as well as other minority people in Thailand, have the rights to receive public support according to the rights outlined in the clause of Thai constitution. Hence, the verification and registration process for the OA in the Chanae district of Narathiwas province was undertaken. All aspects of the quality of life of the study OA are significantly affected due to their migratory living style, which depends on the quantity of natural products harvestable or wild animals caught for adequate consumption. Good livelihood for ending hunger and poverty (SDG 1 and 2), equitable access to health care (SDG 3) and quality education (SDG 4), the three of the seventeen SDGs endorsed by the UN, have been prioritized by Thai official agencies as the primary targets for upgrading the living conditions of all Thais, including the OA in this study. Additionally, reducing inequality (SDG 10) in accessing public support services with the aim for achieving the three targeted SDGs equitably was stressed in this project. The verification and registration process for the OA in the current study was the first and principal action which initiated the cooperation among the related local governmental agencies under the administration of the SBPAC.

The actions performed were based on the thematic principle of establishing equitable accessibility to public support, as other Thai citizens are able, without the significant disruption of the OA’s identity and traditional living. Many previous projects in Thailand and Malaysia involving the resettlement of the OA to new living areas failed because their traditional living styles were abruptly changed due to the policies being implemented without receiving their agreement beforehand. The abrupt changes from traditional to modernized living conditions adversely affected the OA’s traditional ways of life and cultural practices. Most of the resettled OAs soon left the new housing provided by the officials and returned to their previous living sites in the forest. We learned from our previous experiences and were aware that it is necessary to balance the preservation of the identity of this ethnic group and their traditional living style with the officially supported modern living. If the changes were not familiar to the OA, they would reject the offers. Compulsory changes by the provision of certain support programs, despite seemingly useful from the provider’s perspective, commonly bring about conflict or project failure. For these reasons, the SBPAC first conducted integrative actions of local governmental agencies by surveying the OA’s requirements, living styles, cultural beliefs and practices and, especially, their willingness to adopt the registration and development programs provided for the improvement of their well-being. After the legal conditions were fulfilled and the OA’s consent to join the program was obtained, then the integrative actions were started.

The program in this study prioritized the improvements of the OA’s livelihoods, health care and education as initial and urgent targets for receiving local governmental support, because these were considered powerful influencers that interactively affected individuals’ well-being. It was known that the OAs lived by harvesting natural forest products for their daily household consumption. They had no knowledge of how to plant vegetables or other plants or to raise livestock for their food reserves. Normally, they followed a nomadic lifestyle, migrating to a new location in the mountain range every 7–10 days on average, when the harvestable forest products in their current living area became inadequate for consumption. The increased Thai population and accompanying requirements of land for agriculture and forest industries led to ecological changes in the forests. Both natural and man-made ecological changes in the forest have had a negative impact on the amount of forest products harvestable by the OA, resulting in an insecure food supply among the OA. This is the reason why some of the OA were required to permanently settle in a single living location or come down from the hills for labor work in the commercial area of the district. To reduce poverty and lack of food security in response to the first and second SDG goals, our program encouraged and supported the OA to settle permanently in locations along the forest margins, as well as teaching them planting and livestock raising techniques. After the preliminary discussions with the OA and the program were undertaken, we believe that this method of providing social support is suitable for and satisfies the OA very much, in that their ancestral lifestyle and beliefs have not been seriously affected.

The traditional health beliefs among the OA were based on their strong belief in supernatural powers rather than their own inner power or self-efficacy or control in managing their own health. According to the health locus of control concept, they had a lower belief in an internal health locus of control than in an external health locus of control. This kind of belief causes adverse effects to a person’s health [9,10,11,12,13]. Additionally, they lacked the conceptual thoughts or knowledge necessary to generate an appropriate health belief model (HBM) [14] to care for their own health. This concept was also recently used to explain the lack of compliance for receiving a COVID-19 vaccine during the COVID-19 pandemic and vaccination campaign [15,16]. From our interviews with the OA, we learned they were very anxious when discussing modern medical care. Their long-held perception was that attending a hospital led to a dreadful outcome or death. With the help and psychological support from the local public health volunteers, they felt more secure and relaxed. Apart from medical treatment, we believe that the health education from the UHC program will enable them to voluntarily follow health prevention and promotion advice. Previous studies have found that improved knowledge followed by attitudes and practices together influenced soil-transmitted helminth (STH) infection control among the OA in Malaysia [17,18].

The UHC payment scheme in Thailand includes all aspects health services, i.e., health promotion, disease prevention, treatment and rehabilitation, and is available to all Thais from birth to death. Local public health volunteers are the points of first contact in the system when accessing health services. The UHC payment scheme ensures that all Thais will receive the holistic health services equitably. After the OA in this study were successfully registered as Thai citizens, they had equal rights as Thai people to access UHC programs. A study showed that the OA in Malaysia were found to have shorter life expectancies than the Malay people, with overall life expectancies of 53 years (54 years for females and 53 years for males) [19]. Common diseases diagnosed in the OA in Malaysia were STH infections, pulmonary diseases, liver diseases and malnutrition, all of which were occasionally severe enough to cause death [5,20]. We expect that the UHC program will be of considerable benefit for the healthy lives of the OA in this study.

Education is another target of action planned to be promoted in parallel with improving living conditions and health care. Quality education, either formal or informal, can help the OA children or youths to understand Thai or train them vocationally to offer them more choices of career in the future. The OA in this study were encouraged to allow their children of school age to attend formal education, while informal (non-school) as well as vocational education programs were available for adult OA. Because the OA have only their own spoken language but no written language, teaching them the Thai language requires teachers who understand the OA language well. We found that most of the OA parents and children appreciated this offer of educational opportunity.

The small sample size is a limitation of this study. The difficulty of travelling to the OA living areas where they reside on the hilltops and their lifestyle of hunting or harvesting in the forest during daytime are the causes of this issue. However, we made an effort to include all available OA, including their leaders, in our interviews. Male OA of a comparable age range were predominantly included for the interview, since they were the main group of OA making their livings and leading their family members’ lives.

## 5. Conclusions

“We will never leave anyone behind”, a theme of social equality compatible with the UN-endorsed SDGs, was the major concept of the current action plan implemented for the OA communities in this study. We found that the OA in this study were satisfied with the officially provided support. Herein, we suggest that the changing of any indigenous people’s living conditions while aiming for their better well-being by an official project should carefully consider their traditional beliefs and practices. Changing living conditions or implementing obligatory public services that markedly disrupt a minority’s ancestral beliefs and lives and, significantly, without their willingness to adopt these changes as if they are sharing the ownership of the project, commonly result in unfavorable outcomes. Finally, we suggest that the long-term follow-ups of the OA’s accessibility to official services or support programs will elucidate the sustainable benefits and satisfaction among the recipients of the support.

## Figures and Tables

**Figure 1 ijerph-20-05018-f001:**
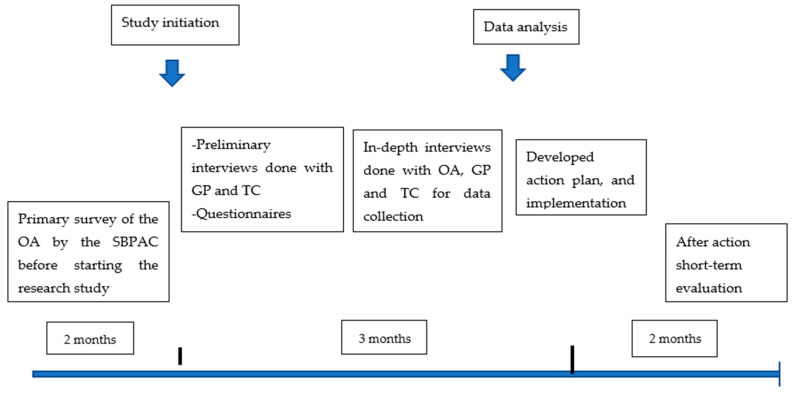
Sequence of activities in this research.

**Table 1 ijerph-20-05018-t001:** Characteristic of the study participants.

	Setting						
	Toapaku Village			Biyis Village			Total
Characteristic	Orang Asli	Government Personnel	Thai Community	Orang Asli	Government Personnel	Thai Community	
Sex							
Female	1	1	0	0	0	0	2
Male	4	2	3	4	2	3	18
Age (Years)							
20–30	0	0	0	0	0	0	0
31–40	1	1	0	1	0	0	3
41–50	3	1	1	1	1	0	7
50+	1	1	2	2	1	3	10
Position							
Head of office or community leader	1	2	2	1	1	2	9
Governmental staff or OA villager	4	1	0	3	1	0	9
Religious leader	0	0	1	0	0	1	2
CODE	(OAa)	(GPa)	(TCa)	(OAb)	(GPb)	(TCb)	

## Data Availability

The study data and analysis methods are described in the Material and Methods section of this paper. No data were deposited in other pre-print servers.

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
