# Peer review of "Developing and Implementing an Action Plan among the “Orang Asali” Minority in Southernmost Thailand for Equitable Accessibility to Public Health Care and Public Services Following the United Nations Sustainable Development Goals"

_ijerph, 2023, doi:10.3390/ijerph20065018_

Round 1
Reviewer 1 Report
This paper uses a qualitative assessment of Orang Asali, an ethnic minority living in Narathiwas province, the southernmost province of Thailand(OA) requirements and barriers to full access to public services were qualitatively assessed. Specifically, they interviewed OA, local government officials and Thai community leaders about the living conditions and health status of 1 7 O A. Finally, the paper puts forward a suggestion, that is, before implementing the corresponding OA social support program, it is necessary to carefully consider the difference between their original lifestyle and their modern lifestyle, so as to guide the adjustment of the plan and the specific implementation process to obtain better results. However, the following issues need to be properly addressed before publishing. Specific comments are as follows:
First of all, the language should be more fluent and smoother, some expressions and wordings are inappropriate or even incorrect. The manuscript should be significantly improved by (1) having a more coherent and better presentation, especially in the introduction and data description and conclusion sections. (2) The citation format of the literature review section should be more in line with the relevant requirements of the submitted manuscript. (3) Proper nouns need to determine whether to use initial capitalization and whether to retain the full name and abbreviation in each noun cited after the abbreviation, also issues to be considered in this manuscript.
Second, in terms of research methodology. The semi-structured research method favors qualitative research, and the set of questions may be misleading, and the manuscript also expresses concern about the problem of "language barrier", of course, it also adopts the way of "asking the translator to paraphrase the questions", but However, it is undeniable that this approach is not a good way to obtain first-hand information, and how to ensure the originality of the information is also a better reference for subsequent descriptive or qualitative research. Suggestions for optimization: In terms of research methods, could the authors add some scale or rating category topics, which would be more intuitive for the analysis of results. It should also be added that in the introduction, the effects achieved in Thailand's original public services and social security for OA should be supplemented with multidimensional and informative data to support and explain the need and importance of the study.
The third is in the content of the study. The authors worked the sample qualitatively and quantitatively, designing questions and conducting interviews in targeted sections. However, certain concerns had to be raised about this, and the specific content of the concerns were first, whether the sample size was selected too small, and second, how to ensure that the age gap would not affect the consistency and robustness of the question set for the group of OA, in other words, whether the credibility of the results was trusted. Third, the questions for the two groups of GP and TC are not differentiated and targeted enough. This is because although both groups, GP and TC, belong to the personnel in the administrative system in the article, their specific identity positioning is different and the relationship with OA is also different. Suggestions: (1) Expand the sample size as well as set the conditions for sample selection, because factors such as status and economic status can affect the results of interviewers' answers to questions. (2) Set more targeted questions for the two groups, GP and TC.
Finally, about the third part of the article - "Post-action short-term assessment". The approach taken in this summary is to trace visits. Suggestions: (1) Whether it is possible to set up scoring questions on OA registration intention, registration convenience, living conditions, health services and education status, etc., so that the data visualization will be stronger, the evaluation will be more convincing, and the development of the project mentioned in the article can be more supportive. Again, this issue also appears in article 3.2.1. Suggestions: (1) Specific official data can be added to supplement the basic information of OA residents in the place.
Reviewer 2 Report
Major Comments:
None
Minor Comments:
1. The abbreviation "TC" is first introduced on line 96. Please clarify that this stands for "Thai community leaders or associated" at this time.
2. I see that the majority of individuals interviewed were male (18/20) and none were under the age of 30. I'm assuming this is because the majority of the village and government leaders were older males, but can the Authors please give some further insight as to why those interviewed were not more representative as far as gender/biological sex and age?
3. Can the Authors please discuss if the OA are absolutely required to accept Thai citizenship in order to access services. If so, please discuss if there would be any merit in giving OA who don't want to be Thai citizens the option of still accessing government services. In many other countries non-citizens can still access basic services such as healthcare.
Author Response
Please see the attached file , Thank you.

Round 2
Reviewer 1 Report
The authors have probably made the following corrections and additions: (i) additional interview questionnaires; (ii) a further response and statement of safeguards against possible information errors arising from oral translations; (iii) formatting and capitalisation issues throughout the text; (iv) an additional statement of the original policy context of living conditions in OA in Thailand; (v) a specific distinction and statement of the potential overlap between GP and TC questions and the potential for classification of personnel (which the authors also acknowledge as a limitation). The authors acknowledge some limitations in this regard, and the amendments are reflected in '2.2 Data collection and preparation'.6 Again, they state that their conclusions are relevant, i.e. they call on the authorities to conduct a long-term assessment after certain events, as opposed to this short-term assessment, in order to focus on and support the survival and welfare conditions of OA. The future development of OA. It is also worth noting that the response is very detailed and sincere, and the author's care and modesty is felt. My recommendation is to accept.